# Latent classes of unhealthy behaviours and their associations with subsequent sickness absence: a prospective register-linkage study among Finnish young and early midlife employees

Jatta Salmela ,[1] Jouni Lahti,[1] Noora Kanerva,[2] Ossi Rahkonen ,[1] Anne Kouvonen ,[3,4] Tea Lallukka [1]

¹Department of Public Health, Faculty of Medicine, University of Helsinki, Helsinki, Finland
²Department of Food and Nutrition, Faculty of Agriculture and Forestry, University of Helsinki, Helsinki, Finland
³Faculty of Social Sciences, University of Helsinki, Helsinki, Finland
⁴Centre for Public Health, Queen's University Belfast, Belfast, UK

**Correspondence to**
Dr Jatta Salmela;
jatta.salmela@helsinki.fi

## ABSTRACT

**Objectives** Unhealthy behaviours are associated with increased sickness absence (SA), but few studies have considered person-oriented approach in these associations. Using latent class analysis, we examined clustering of unhealthy behaviours among Finnish municipal employees and their associations with subsequent SA.

**Design** A prospective register-linkage study.

**Setting** Unhealthy behaviours (low leisure-time physical activity, non-daily fruit and vegetable consumption, insufficient sleep, excessive alcohol use and tobacco use) were derived from the Helsinki Health Study questionnaire survey, collected in 2017 among 19- to 39-year-old employees of the City of Helsinki, Finland.

**Participants** A total of 4002 employees (81% women) of the City of Helsinki, Finland.

**Primary outcome measures** The questionnaire data were prospectively linked to employer's SA register through March 2020. Associations between latent classes of unhealthy behaviours and subsequent SA (1–7 days/8+ days/all lengths) were examined using negative binomial regression.

**Results** Among women, a three-class latent class model was selected: (1) few unhealthy behaviours (84%), (2) excessive alcohol and tobacco use (12%) and (3) several unhealthy behaviours (5%). Women belonging to classes 2 and 3 had increased SA rates compared with those in class 1, regardless of the length of SA spells. Among men, a 2-latent class model was selected: (1) few unhealthy behaviours (53%) and (2) several unhealthy behaviours (47%). Men belonging to class 2 had increased rates of 1–7 days' SA compared with men in class 1.

**Conclusions** This study suggests that preventive actions aiming to reduce employees' SA should consider simultaneously several unhealthy behaviours. Targeted interventions may benefit of identifying the clustering of these behaviours among occupational groups.

## INTRODUCTION

Health behaviours have a major contribution to employees' sickness absence (SA). It is estimated that 15%–31% of SA could

## STRENGTHS AND LIMITATIONS OF THIS STUDY

⇒ Applying person-oriented approach enabled us to identify unobserved population groups that share similar patterns of unhealthy behaviours.
⇒ We could link questionnaire data on employees' health behaviours to employers' register data on different lengths of sickness absence (1–7 days, 8+ days and all lengths) with a mean follow-up time of 2.13 years.
⇒ Self-reported measures of health behaviours may be biased, which may influence the identified latent classes of unhealthy behaviours.
⇒ Although the large proportion of women well represents the gender distribution in the target population and in the municipal sector in Finland in general, the small number of men limits the interpretation of the findings among men and the gender comparisons.

be attributed to unhealthy behaviours.[1 2] In addition to their independent contributions, health behaviours can mediate some of the effects of working conditions and socioeconomic circumstances on SA: for instance, unhealthy behaviours (eg, smoking and binge drinking) may be used to cope with stressful working conditions.[3 4] In Finland, as in most high-income countries, the leading causes for medically certified SA are mental and musculoskeletal disorders.[5 6] Unhealthy behaviours, such as low physical activity, poor sleep, binge drinking and smoking, have been associated with both medically certified SA and self-certified SA.[1 7–9] However, the results are not fully consistent.[2 10] Diet appears to have a minor contribution to SA,[2 3 7 11] but since obesity is consistently associated with SA,[12] dietary aspects—which play a key role in weight management—should not be neglected. The possible mechanisms

and pathways between unhealthy behaviours and SA have been suggested to proceed, for instance, through increased risk for chronic diseases, risk-taking lifestyle and decreased immune system (leading to, eg, common cold).[1–3] Additionally, working conditions and socioeconomic circumstances may explain some of the associations.[3]

Accumulation of several unhealthy behaviours has been shown to increase SA more than individual unhealthy behaviours.[2 7 13] Our previous study on midlife and ageing Finnish employees found that the joint contribution of physical inactivity and smoking was especially detrimental for employer's cost of 1–14 days' SA.[13] Health behaviours tend to be clustered within population groups,[14 15] and these clusters may have synergistic effects on health.[14] Considering clustering of unhealthy behaviours can help policymakers and researchers to design targeted interventions to improve employees' health behaviours and reduce SA. However, to the best of our knowledge, no studies have examined how clustering of unhealthy behaviours is associated with SA. Clustering techniques, such as latent class analyses, can provide more holistic approach on how health behaviours contribute to SA compared with summary indices[2 7] that consider each risk factor equally and disregard their interconnections.[16]

This study aimed to identify latent classes of five unhealthy behaviours among 19- to 39-year-old employees of the City of Helsinki, Finland. Furthermore, using linkage to employer's SA register, we aimed to examine the associations between the latent classes with subsequent SA.

## METHODS
### Data and study population
This study is a part of the Helsinki Health Study of young and early midlife employees of the City of Helsinki.[17] The City of Helsinki is the largest employer in Finland with around 38 000 employees and hundreds of occupational titles. The target population included 11 459 employees who were born in 1978 or later, who had a job contract of at least 50% of regular work hours per week and whose employment contract had lasted at least 4 months before the data collection began in autumn 2017. Data were collected via online and mailed questionnaires, which included a large variety of questions related to participants' social and economic characteristics and health behaviours. Additionally, shorter telephone interviews were conducted to target those who did not respond online or via email. The overall response rate was 51.5% (n=5898).[17] The survey data were linked to employer's personnel register data for those who gave their written informed consent (82% of respondents, n=4864). We excluded telephone interviewees (n=651) since the interviews did not include all the variables of interest in this study, as well as participants who had missing data on working time or on all health behaviours of interest (n=34), or who had extreme values in health behaviours

(n=177) (online supplemental file 1, figure S1). The final analytical sample included 4002 participants (81% women).

### Health behaviour measures
We included five unhealthy behaviours from the survey: (1) low leisure-time physical activity (LTPA), (2) non-daily fruit and vegetable (F&V) consumption, (3) insufficient sleep, (4) excessive alcohol use and (5) tobacco use (see online supplemental file 2). Since it is not computationally possible to include too many multi-categorical variables or variables with very small group sizes in the latent class analysis (LCA) models, we dichotomised all health behaviour measures taking into consideration current guidelines and group sizes in the variables. Participants were inquired about their weekly volume and intensity of exercise in their leisure time or while commuting during the past 12 months. Four levels of intensity were provided, and they were multiplied by the time used per week in LTPA, yielding weekly metabolic equivalent task (MET)-hours.[7] Then, we dichotomised participants to those with high/moderate LTPA and those with low LTPA by using a cut-point of 20 MET-hours. Twenty MET-hours equals, for instance, 2.5 hours brisk walking and 1.5 hours walking, which was considered closely to correspond current guidelines.[18 19]

F&V consumption during the past 4 weeks was inquired using a 14-item food frequency questionnaire. We dichotomised participants into daily (once a day or more F or V) and non-daily F&V consumers. Subjective experience of sleep was used as a sleep measure. We dichotomised participants into those who estimated that they sleep always/often sufficiently and those who estimated that they sleep seldom/never sufficiently. Alcohol use combined the measures of total weekly alcohol use and binge drinking behaviour. Weekly alcohol use was calculated based on participants' estimation on how often they consume different alcohol types (beer/cider, wine and spirits). Seven frequency alternatives were provided for each question, with one unit of alcohol equalling 12 g ethanol. Based on the Finnish Current Care Guidelines on alcohol consumption,[20] 7 weekly units for women and 14 weekly units for men (ie, moderate risk levels) were considered as cut-points. Additionally, participants were asked how often they drink six units or more at once (six response alternatives). We dichotomised those drinking less than 7/14 (women/men) units per week and binge drinking less than once a month into moderate alcohol users, and others to excessive alcohol users. We merged those not drinking alcohol at all (4% of women and 2% of men) with moderate alcohol users since their associations with SA did not differ from those drinking moderately alcohol. Participants were provided four alternatives to estimate their use of tobacco products (cigarettes, e-cigarettes and snus): 'yes, daily', 'sporadically', 'not nowadays' and 'never'. We dichotomised participants into never/ex-users, and those using daily/occasionally tobacco products.

## SA measures

The data on SA were derived from the personnel register of the City of Helsinki. The follow-up of SA began 1 day after receiving the completed survey questionnaire and continued until 31 March 2020 or until the end of one's employment contract, whichever came first. The time limit was selected so that we could exclude the potential influence of the COVID-19 pandemic to the results. The mean follow-up time was 2.13 years. We combined overlapping and consecutive SA spells and divided them into SA spells of 1–7 days and 8+ days. During the follow-up, the City of Helsinki had a policy that 1–7 days' SA could be given to an employee by their supervisor, nurse, occupational physiotherapist or physician, whereas 8+ days' SA required a medical certification approved by a physician. The policy was the same for all employees. Additionally, we analysed all lengths' SA.

## Covariates

We stratified all analyses by gender (woman/man), given that notable gender differences have been observed in SA and health behaviours,[21 22] and clustering of health behaviours may vary by gender.[15] Age included categories of 19–29, 30–34 and 35–39 years. Marital status was derived from the questionnaire and was dichotomised into married/cohabiting and other. In the questionnaire, participants were inquired whether they had any 0 to 18-year-old children living in their household ('yes/no'). Occupational class was derived from the employer's personnel register for those who gave their informed consent for register linkage (82%), and for others, the information was derived from the questionnaire. Occupational class included four groups: managers and professionals (eg, teachers and physicians), semi-professionals (eg, nurses and foremen), routine non-manual workers (eg, childcare and elderly care workers) and manual workers (eg, care assistants). It is noteworthy that in recent years the City of Helsinki has outsourced most of their manual work (eg, cleaning and transport work), and therefore the proportion of manual workers employed by the city is now very low. Prior SA, especially past year's SA, is known to predict future SA.[10 23] Thus, we included prior SA of any length during 1 year before participant's response to the questionnaire.

## Statistical methods

We first tabulated descriptive statistics by key exposure variables. Then, incidence of SA days per 10 person-years were calculated by individual health behaviours using negative binomial regression. We identified latent classes of unhealthy behaviours using LCA. LCA is a person-oriented statistical procedure to detect latent (unobserved) subgroups, which share certain outward characteristics, within a heterogeneous population.[24 25] This subtype of structural equation modelling uses categorical indicator variables to form latent classes based on the indicator variables. Participants are assigned to the latent classes based on their probability of class membership. We used the following statistical criteria for selecting the most optimal number of latent classes: Bayesian information criterion, Akaike information criterion, average posterior probabilities of class membership (>0.8), class sizes (>50 cases or >5% of the sample) and entropy (>0.8).[25] One-class to five-class models were run, and the model fit evaluation process is shown in online supplemental file 1, table S1. Additionally, we considered the interpretability of the models to select the final models.[25]

We used negative binomial regression to examine associations between latent classes of unhealthy behaviours and subsequent SA due to overdispersion in the data. Rate ratios and predictive margins with 95% CIs were calculated. Model 1 was adjusted for age, and model 2 further for marital status, children living in the household, occupational class and prior SA. Natural logarithm of the follow-up time was included as an offset variable in all models to consider differences in the follow-up times between participants. All analyses were performed using STATA V.17.0 (StataCorp).

## Patient and public involvement

Patients or the public were not involved in this study.

## RESULTS

### Characteristics of study population

Most participants had at least one unhealthy behaviour (67% of women and 83% of men), whereas under 1% of women and men had all five unhealthy behaviours. Low LTPA and insufficient sleep were equally common among women and men (table 1). However, non-daily F&V consumption, excessive alcohol use and tobacco use were more common among men than among women. Most women and men were married/cohabiting and around 40% had children living in their household. Only 3% of women were manual workers while the corresponding proportion for men was 13%.

During the follow-up, we recorded altogether 117 SA days/10 person-years for women and 93 SA days/10 person-years for men. Of women, 15% had no 1–7 days' SA, 69% had no 8+ days' SA, and 18% had no SA of any length during the follow-up. For men, the corresponding figures were 18%, 75% and 17%. Participants with healthier behaviours had less SA than those with unhealthier behaviours in general (table 2). However, F&V consumption and alcohol use were exceptions among men in terms of 8+ days' SA: those with healthier behaviour had more or equally 8+ days' SA compared with those with unhealthier behaviour. When scrutinising all lengths' SA, the greatest differences between healthy and unhealthy behaviour groups were seen in tobacco use among women and in sleep among men.

### Latent classes of unhealthy behaviours

The most optimal number of latent classes of unhealthy behaviours was three for women and two for men (figure 1

**Table 1** Characteristics of the participants by sociodemographic factors and health behaviours among women and men

| | Women (n, %) | Men (n, %) |
|---|---|---|
| **Total** | **3228 (80.7)** | **774 (19.3)** |
| **Health behaviours** | | |
| Leisure-time physical activity* | | |
| High or moderate activity | 2689 (84.4) | 651 (85.3) |
| Low activity | 499 (15.7) | 112 (14.7) |
| Fruit and vegetable consumption | | |
| Daily | 2595 (80.5) | 463 (60.0) |
| Non-daily | 629 (19.5) | 309 (40.0) |
| Sleep sufficiency | | |
| Mostly sufficient sleep | 2146 (66.9) | 521 (67.8) |
| Insufficient sleep | 1064 (33.2) | 248 (32.3) |
| Alcohol use† | | |
| Moderate | 2492 (79.9) | 423 (55.8) |
| Excessive | 626 (20.1) | 335 (44.2) |
| Tobacco use‡ | | |
| No | 2430 (75.8) | 471 (61.1) |
| Currently or occasionally | 777 (24.2) | 300 (38.9) |
| **Sociodemographic factors** | | |
| Age | | |
| 19–29 years | 1049 (32.5) | 197 (25.5) |
| 30–34 years | 1108 (34.3) | 252 (32.6) |
| 35–39 years | 1071 (33.2) | 325 (42.0) |
| Marital status | | |
| Married or cohabiting | 2122 (65.7) | 570 (73.6) |
| Other | 1106 (34.3) | 204 (26.4) |
| Children living in the household | | |
| No | 1851 (57.3) | 467 (60.3) |
| Yes | 1377 (42.7) | 307 (39.7) |
| Occupational class | | |
| Managers and professionals | 895 (27.7) | 241 (31.1) |
| Semi-professionals | 1402 (43.4) | 242 (31.3) |
| Routine non-manual workers | 843 (26.1) | 191 (24.7) |
| Manual workers | 88 (2.7) | 100 (12.9) |

*Leisure-time physical activity (LTPA) included physical activity during leisure time and active commuting. High or moderate LTPA was considered as ≥20 metabolic equivalent task (MET)-hours per week and low LTPA as <20 MET-hours per week.
†Moderate alcohol use: ≤7 units of alcohol per month and binge drinking less than once a month among women, and ≤14 units of alcohol per month and binge drinking less than once a month among men. Excessive alcohol use: >7 units of alcohol per month and binge drinking less than once a month among women, and >14 units of alcohol per month and binge drinking less than once a month among men.
‡Tobacco use included the use of cigarettes, e-cigarettes and snus.

and online supplemental file 1, figure S1. Although model fit statistics preferred the two-class solution for women, three classes were selected as they were interpretatively reasonable and provided new information about the data.

Most statistical criteria preferred the three-class solution for men, but we selected two classes due to one too small class size (n=39) in the three-class solution. Marginal means for each unhealthy behaviour within latent classes are shown in online supplemental file 1, table S2.

Of women, 84% had the highest posterior probability for belonging to class 1, and for classes 2 and 3, the corresponding proportions were 12% and 5% (figure 1A). Class 1 was characterised by overall low probabilities of having unhealthy behaviours. Class 2 was characterised especially by excessive alcohol use and tobacco use, whereas probabilities for other unhealthy behaviours were somewhat low. In class 3, there were increased probabilities for all other unhealthy behaviours except excessive alcohol use. Of men, 53% had the highest posterior probability for belonging to class 1, and 47% to class 2 (figure 1A). Class 1 was characterised by somewhat low probabilities of having any unhealthy behaviours. The probabilities of having unhealthy behaviours were overall increased in class 2, and it was especially characterised by low LTPA, non-daily F&V consumption and excessive alcohol use.

### Associations between latent classes of unhealthy behaviours and SA

Women belonging to classes 2 and 3 had increased SA rates compared with class 1 (table 3). However, the associations with 8+ days' SA were not statistically significant. Women belonging to classes 2 and 3 had increased rates of 1–7 days' SA even after adjustment for age, marital status, children living in the household, occupational class and prior SA (table 3, M2). Men belonging to class 2 had increased SA rates compared with class 1 (table 4). However, statistically significant association was found only for 1–7 days' SA in the age-adjusted model (table 4, M1). This association attenuated after further adjustments (table 4, M2), especially after adjustment for occupational class.

### DISCUSSION
### Summary of the main findings
By using the LCA method, we selected three latent classes of unhealthy behaviours among women, characterised as follows: (1) few unhealthy behaviours, (2) excessive alcohol use and tobacco use and (3) several unhealthy behaviours. Among men, we selected two latent classes with the following characteristics: (1) few unhealthy behaviours and (2) several unhealthy behaviours. Women in classes 2 and 3, and men in class 2 had increased rates of 1–7 days' SA compared with class 1. The associations between latent classes of unhealthy behaviours and 8+ days' SA were not statistically significant either among women or men.

### Comparisons to the previous literature
The majority of women and men were most likely to belong to class 1, characterised by overall healthier behaviours. Similarly, a systematic review of the clustering of smoking, nutrition, alcohol and physical activity in

**Table 2** Incidence of sickness absence (SA) days per 10 person-years with 95% CIs (in parenthesis), by health behaviours among women and men

| Health behaviours | Women (n=3228) | | | Men (n=774) | | |
|---|---|---|---|---|---|---|
| | 1–7 days' SA | 8+ days' SA | All lengths' SA | 1–7 days' SA | 8+ days' SA | All lengths' SA |
| Leisure-time physical activity* | | | | | | |
| High or moderate activity | 62 (59 to 64) | 67 (58 to 77) | 128 (122 to 135) | 50 (46 to 55) | 40 (30 to 55) | 90 (81 to 101) |
| Low activity | 72 (66 to 80) | 74 (55 to 99) | 146 (130 to 163) | 64 (51 to 81) | 52 (26 to 104) | 116 (89 to 152) |
| Fruit and vegetable consumption | | | | | | |
| Daily | 61 (58 to 63) | 67 (58 to 78) | 127 (121 to 134) | 47 (42 to 52) | 45 (31 to 65) | 92 (81 to 105) |
| Non-daily | 74 (68 to 81) | 74 (57 to 96) | 148 (133 to 163) | 59 (52 to 68) | 36 (24 to 56) | 96 (83 to 112) |
| Sleep sufficiency | | | | | | |
| Mostly sufficient sleep | 61 (58 to 64) | 59 (51 to 70) | 120 (113 to 127) | 46 (41 to 51) | 32 (22 to 46) | 77 (69 to 88) |
| Insufficient sleep | 68 (63 to 73) | 88 (72 to 108) | 155 (143 to 169) | 64 (56 to 74) | 62 (40 to 95) | 127 (108 to 149) |
| Alcohol use† | | | | | | |
| Moderate | 60 (58 to 63) | 66 (57 to 76) | 125 (119 to 132) | 49 (44 to 55) | 42 (29 to 61) | 91 (80 to 104) |
| Excessive | 75 (69 to 82) | 81 (62 to 105) | 156 (140 to 172) | 56 (50 to 64) | 42 (27 to 66) | 99 (85 to 115) |
| Tobacco use‡ | | | | | | |
| No | 58 (56 to 61) | 62 (53 to 72) | 119 (113 to 126) | 48 (43 to 53) | 37 (26 to 53) | 84 (74 to 96) |
| Currently or occasionally | 79 (74 to 85) | 91 (72 to 114) | 169 (155 to 185) | 59 (52 to 67) | 49 (32 to 77) | 108 (93 to 126) |

*Leisure-time physical activity (LTPA) included physical activity during leisure time and active commuting. High or moderate LTPA was considered as ≥20 metabolic equivalent task (MET)-hours per week and low LTPA as <20 MET-hours per week.
†Moderate alcohol use: ≤7 units of alcohol per month and binge drinking less than once a month among women, and ≤14 units of alcohol per month and binge drinking less than once a month among men. Excessive alcohol use: >7 units of alcohol per month and binge drinking less than once a month among women, and >14 units of alcohol per month and binge drinking less than once a month among men.
‡Tobacco use included the use of cigarettes, e-cigarettes and snus.

adults found that a majority of included studies reported a 'healthy' cluster, characterised by the absence of any unhealthy behaviours.[14] This was not affected by in how health behaviours were defined or by the used clustering analysis method.[14] Some more recent studies have also identified a class of overall healthier behaviours.[26–28] Additionally, previous studies have found especially alcohol consumption and smoking often clustering,[14 15] which we also observed in women in class 2. However, in men, this was not observed with two latent classes. Further analyses revealed that with a three-class solution in men, clustering of excessive alcohol use and tobacco use existed similarly as in women. Clustering of low LTPA and non-daily F&V consumption, which we observed in class 2 among men, has been found in many of the previous studies.[14 15] However, Noble et al's systematic review did not find clustering of physical inactivity, poor diet and excess alcohol use—the combination that we found to reflect class 2 in men—in any of the included studies.[14] Finally, clustering of several unhealthy behaviours have been observed in many previous studies,[14 26] which we also could observe in class 3 among women and class 2 among men.

To our knowledge, no previous studies have examined associations between latent classes of unhealthy behaviours and SA, although the relationship between health behaviours and SA are broadly studied in general. Concerning single unhealthy behaviours, previous studies have associated low LTPA,[1 9 29 30] poor sleep,[31 32] excessive alcohol use[1 33] and smoking[1 2 29 30 32 34] with SA, while the

contribution of poor diet to SA has been modest.[2 3 7 11 29] Although diet has not been associated with SA as strongly as other health behaviours, we found that inadequate F&V consumption was one major characteristic of class 3 among women and class 2 among men—the classes that were associated with increased subsequent SA. Our previous study on midlife and older employees also showed that the joint contribution of F&V consumption and LTPA to SA might be stronger than the individual contribution of LTPA.[11] However, since F&V consumption reflects only partially participants' overall diet, further studies that consider dietary patterns more comprehensively are needed.

Our previous study showed that midlife and older employees with three or more unhealthy behaviours had higher cost of 1–14 days' SA than employees without any unhealthy behaviours.[7] In particular, low LTPA, poor sleep and current smoking increased the SA cost.[7] Another study by our research group found that the joint contributions of low LTPA, poor sleep and smoking to 1–14 days' SA cost were stronger than the contributions of these health behaviours individually.[13] A Norwegian study on general working population found that an exposure to multiple health-related risk factors (low physical activity, unhealthy diet, obesity and current smoking) was associated with increased subsequent 1–14 days' and 15+ days' SA.[2] Additionally, a Danish study on private sector employees found that exposure to multiple health-related risk factors (dyssomnia, overweight, unhealthy food

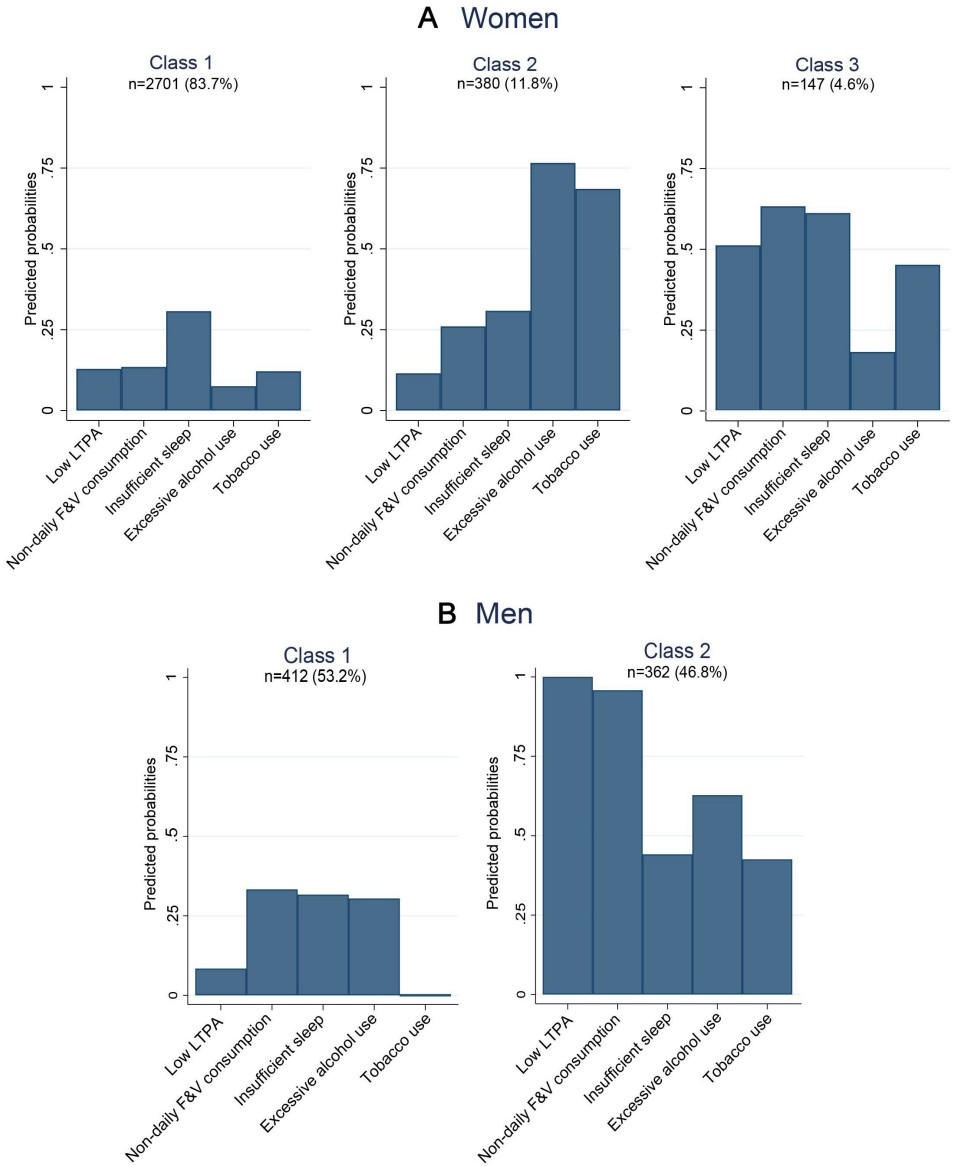

**Figure 1** Latent classes of unhealthy behaviours among women (A) and men (B). F&V, fruit and vegetable; LTPA, leisure-time physical activity.

habits, smoking, excessive alcohol use and low physical activity) were related to increased 1–14 days' SA.[32] These previous findings are concordant with our finding which indicated that the latent classes of several unhealthy behaviours (class 3 for women and class 2 for men) were related to increased SA rates.

We found that latent classes of unhealthy behaviours were associated with 1–7 days' SA among women and men, but not statistically significantly with 8+ days' SA. In contrast, previous studies have found stronger associations for longer SA spells.[2 3] There is some evidence that younger employees have more often short-term SA and older employees long-term SA,[34 35] which may partly explain our findings. Another explanation is that the follow-up period of 2.13 years may not be long enough to ensure the associations with 8+ days' SA since their rate during the follow-up was relatively low.

Previous research has shown that clustering of unhealthy behaviours is strongly related to socioeconomic position.[14 15 26 27 36] Similarly, we found that managers and professionals were more likely to belong to the 'healthiest' latent classes (online supplemental file 1, table S3). However, occupational class together with other sociodemographic factors explained only some of the associations between the latent classes of unhealthy behaviours and SA. Since socioeconomic differences in SA are visible already among young employees[37] and employees in the lower socioeconomic positions are more likely to have adverse working conditions (eg, higher exposure to physical workload) that are strongly related to increased SA,[29 34 38 39] these factors should not be neglected when designing targeted health behaviour interventions at workplaces. Burdorf and Robroek have suggested that preventive interventions should simultaneously

**Table 3** Associations between latent classes of unhealthy behaviours and sickness absence (SA) among women

| Latent class | 1–7 days' SA | | 8+ days' SA | | All lengths' SA | |
| --- | --- | --- | --- | --- | --- | --- |
| | M1: adjusted for age | M2: M1+marital status, children living in the household, occupational class, prior SA† | M1: adjusted for age | M2: M1+marital status, children living in the household, occupational class, prior SA† | M1: adjusted for age | M2: M1+marital status, children living in the household, occupational class, prior SA† |
| | RR (95% CI) | | | | | |
| Class 1 | ref. | ref. | ref. | ref. | ref. | ref. |
| Class 2 | 1.39 (1.24 to 1.57) | 1.21 (1.08 to 1.36) | 1.37 (0.92 to 2.02) | 1.35 (0.92 to 1.97) | 1.39 (1.20 to 1.61) | 1.29 (1.13 to 1.48) |
| Class 3 | 1.37 (1.14 to 1.64) | 1.19 (1.00 to 1.42) | 1.31 (0.72 to 2.38) | 1.22 (0.68 to 2.18) | 1.34 (1.07 to 1.67) | 1.18 (0.96 to 1.46) |
| | Predictive margins (95% CI) | | | | | |
| Class 1 | 12.6 (12.1 to 13.2) | 12.2 (11.7 to 12.7) | 13.7 (11.8 to 15.6) | 10.8 (9.3 to 12.2) | 26.2 (24.8 to 27.5) | 23.0 (21.9 to 24.1) |
| Class 2 | 17.6 (15.6 to 19.6) | 14.8 (13.2 to 16.4) | 18.7 (11.8 to 25.5) | 14.5 (9.3 to 19.7) | 36.5 (31.5 to 41.5) | 29.7 (25.9 to 33.5) |
| Class 3 | 17.3 (14.2 to 20.3) | 14.5 (12.1 to 17.0) | 17.9 (7.4 to 28.3) | 13.1 (5.7 to 20.5) | 35.0 (27.4 to 42.6) | 27.2 (21.7 to 32.7) |

Rate ratios (RR) and predictive margins with 95% CIs from negative binomial regression models* are shown.
*Natural logarithm of the follow-up time is included in the models as an offset variable.
†Prior sickness absence of all lengths 1 year before the follow-up, divided by the working time in years during the 1-year's period.

consider improvements in working conditions and health behaviours, and they should be targeted to high-risk and low-educated population groups.[40] These could include, for example, reducing physical and psychosocial strenuousness of work while making healthy choices more easily available, for instance, by supporting active commuting, providing exercise facilities, improving availability of staff canteens providing healthy meals and improving accessibility to occupational health services. Identifying occupational groups among whom these conditions are insufficient and among whom unhealthy behaviours are common is crucial for employers. Additionally, given that younger age predisposes to clustering of unhealthy behaviours (online supplemental file 1, table S3)—and[14 26 27] that health behaviours may be more difficult to modify the older individuals are—preventive actions are especially needed among young employees in the lower socioeconomic positions.

## Limitations and strengths

This study has a few limitations that should be considered. First, health behaviours were self-reported, thus biased estimates are possible. Second, the used cut-points in the health behaviour measures may have affected the identified latent classes. We tested various options and made the final decisions of the dichotomisations based on their consistency with the current guidelines and their proportions in the data. Third, the used cut-point in SA (1–7/8 days) complicates the comparisons to other studies since many previous studies have used cut-points of 3/4 days or 14/15 days to distinguish short-term SA from long-term SA. However, 15+ days' SA were rare in this study population, and the changes made in the SA practices by the City of Helsinki during the follow-up period supported using the chosen cut-point. SA spells of 8+ days were still relatively rare in the study population, and a longer

**Table 4** Associations between latent classes of unhealthy behaviours and sickness absence (SA) among men

| Latent class | 1–7 SA days | | 8+ SA days | | SA days of all lengths | |
| --- | --- | --- | --- | --- | --- | --- |
| | M1: adjusted for age | M2: M1+marital status, children living in the household, occupational class, prior SA† | M1: adjusted for age | M2: M1+marital status, children living in the household, occupational class, prior SA† | M1: adjusted for age | M2: M1+marital status, children living in the household, occupational class, prior SA† |
| | RR (95% CI) | | | | | |
| Class 1 | ref. | ref. | ref. | ref. | ref. | ref. |
| Class 2 | 1.23 (1.04 to 1.45) | 1.11 (0.95 to 1.31) | 1.16 (0.65 to 2.06) | 1.01 (0.58 to 1.77) | 1.20 (0.98 to 1.46) | 1.06 (0.88 to 1.28) |
| | Predictive margins (95% CI) | | | | | |
| Class 1 | 10.0 (8.8 to 11.1) | 9.5 (8.5 to 10.6) | 8.2 (5.0 to 11.4) | 6.3 (4.0 to 8.7) | 18.2 (15.7 to 20.7) | 16.2 (14.2 to 18.2) |
| Class 2 | 12.2 (10.8 to 13.7) | 10.6 (9.4 to 11.8) | 9.6 (5.6 to 13.5) | 6.4 (3.8 to 8.9) | 21.8 (18.7 to 25.0) | 17.2 (14.9 to 19.5) |

Rate ratios (RR) and predictive margins with 95% CIs from negative binomial regression models* are shown.
*Natural logarithm of the follow-up time is included in the models as an offset variable.
†Prior sickness absence of all lengths 1 year before the follow-up, divided by the working time in years during the 1-year's period.

follow-up period could have strengthened the interpretation of the findings. Fourth, the small number of men limits the interpretation of the findings among men and the gender comparisons. The large proportion of women well represents, however, the gender distribution in the target population and in the municipal sector in Finland in general.

Fifth, missing data and non-participation may affect the findings. LCA uses maximum likelihood estimation and assumes missingness at random,[25] thus missing data on health behaviours were allowed. However, we have carefully examined the representativeness of the data and found them to satisfactorily represent the target population (N=11 459).[17] The response rate to the survey was moderate (51.5%), and the non-respondents were somewhat more often men, manual workers, had lower income and had more 15+ days' SA.[17] Additionally, the participants included in this current study were more often of higher occupational class (online supplemental file 1, table S4); thus, our results may be slightly conservative.[26 36] However, the sensitivity analyses showed that the final analytical sample (n=4002) highly resembled the full sample (n=5898) in terms of health behaviours and socioeconomic characteristics (online supplemental file 1, table S4). In addition to the use of comprehensive survey data, a further strength of this study is that we could link the questionnaire survey to employer's SA registers, which is rarely possible. Furthermore, using the person-oriented LCA method to deepen our understanding on the associations between unhealthy behaviours and SA is a novel approach in this study area.

## Conclusions

This study identified three latent classes of unhealthy behaviours for women and two for men. The 'healthiest' classes among women and men showed the lowest SA rates. The associations of the latent classes of unhealthy behaviours were stronger with 1–7 days' than with 8+ days' SA. Thus, by considering the clustering of unhealthy behaviours among young and early midlife employees and intervening in them may reduce employees' short-term SA at least. Occupational class together with other sociodemographic factors explained some of the found associations, thus special focus on employees with lower occupational positions is needed.

**Acknowledgements** The authors thank all participating employees and the personnel administration of the City of Helsinki.

**Contributors** JS was the primary author of the paper, performed the statistical analyses and is responsible for the overall content as guarantor. TL contributed to the study design. JL, NK, OR, AK and TL contributed to the interpretation of the findings. JS, JL, NK, OR, AK and TL critically reviewed the manuscript and approved the final version of the manuscript.

**Funding** JS and TL were supported by the Social Insurance Institution of Finland (Kela) (grant 29/26/2020). OR was supported by the Juho Vainio Foundation (grant 202300041). AK was supported by the Economic and Social Research Council (ESRC) (grant ES/S00744X/1).

**Competing interests** None declared.

**Patient and public involvement** Patients and/or the public were not involved in the design, or conduct, or reporting, or dissemination plans of this research.

**Patient consent for publication** Not applicable.

**Ethics approval** The Helsinki Health Study protocol has been approved by the ethics committees of Department of Public Health at the University of Helsinki (initially 30 November 1998, updated 14 February 2017) and the health authorities of the City of Helsinki (initially 5 October 1999, updated 16 April 2017). The permission to have access to the employer's personnel register data was obtained from the City of Helsinki. Department of Public Health gave an approval (positive statement) for the study, and because the study is observational, ethical approval was not required. The City of Helsinki admitted an ethical approval without a code. Appropriate ethical aspects have been followed in all phases of the study, according to the Declaration of Helsinki.

**Provenance and peer review** Not commissioned; externally peer-reviewed.

**Data availability statement** Data are available upon reasonable request. The Helsinki Health Study survey data cannot be made publicly available due to strict data protection laws and regulations. The data can only be used for scientific research and to the research group's cooperation partners with a reasonable request and study plan. More information on the availability of the survey data can be inquired from the Helsinki Health Study research group (kttl-hhs@helsinki.fi). Register data cannot be shared.

**ORCID iDs**
Jatta Salmela http://orcid.org/0000-0001-7880-834X
Ossi Rahkonen http://orcid.org/0000-0002-7202-3274
Anne Kouvonen http://orcid.org/0000-0001-6997-8312
Tea Lallukka http://orcid.org/0000-0003-3841-3129

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
