## [Reviewer comments · BMJ Open]

This paper was submitted to a another journal from BMJ but declined for publication following peer review. The authors addressed the reviewers' comments and submitted the revised paper to BMJ Open. The paper was subsequently accepted for publication at BMJ Open.

ARTICLE DETAILS

TITLE (PROVISIONAL)	Latent classes of unhealthy behaviours and their associations with subsequent sickness absence: a prospective register-linkage study among Finnish young and early midlife employees
AUTHORS	Salmela, Jatta; Lahti, Jouni; Kanerva, Noora; Rahkonen, Ossi; Kouvonen, Anne; Lallukka, Tea

VERSION 1 – REVIEW

REVIEWER	Åkerström, Magnus Institute of Stress Medicine
REVIEW RETURNED	09-Jan-2023

GENERAL COMMENTS	Thank you for the opportunity to review this manuscript. The study aims to investigate the association between unhealthy behavior and sickness absence among municipality employees of Helsinki using a person-centered approach with latent-class analysis. The study is well executed, and the manuscript is well written. However, there are some aspects concerning the mechanisms studied and the association between unhealthy behavior, employee sickness absence and the working conditions which warrants some further efforts and discussion. Firstly, the authors should preferably discuss the thought mechanism between unhealthy behavior and sickness absence in the introduction and/or the methods section to make the rationale for the study clearer and strengthen the conclusion that sickness absence may be prevented by targeted interventions aiming at increasing unhealthy behaviors. The authors state in the introduction that unhealthy behavior is a major contributor of the employee's sickness absence. It further state that the leading causes for sickness absence is mental and musculoskeletal disorders. What is the direction of these associations? Will unhealthy behavior result directly in sickness absence, or will it result in mental and musculoskeletal disorders which ultimately will result in sickness absence? Or may mental and musculoskeletal disorders, for instance caused by adverse working conditions, result in unhealthy behavior such as low physical activity and alcohol consumption? This need to be clearer to assist the reader in interpreting the outcomes of this study.
--

	Secondly, and in connection to above, the association between unhealthy behavior, employee sickness absence and the working conditions need to be discussed some further. In the introduction, the authors state that unhealthy behavior can mediate the effects of working conditions and socioeconomic circumstances on SA however, but as far as I understand, the references used have shown no effect of working conditions on the association between unhealthy behavior and sickness absence and that working conditions and life-style factors may partly explain the association between education and sick leave. The manuscript would also benefit from a more detailed description of the professions of the study population and how they correspond to the occupational classes and/or the distribution of professions within the different classes as this may give an increased insight in the working conditions within the different classes that may affect the conclusions of the study. In the analyses, the authors control for occupational class which seems to be a measure of socioeconomic position but are there other potential confounder connected to different occupations? Are there factors within the included professions that may affect the measured unhealthy behaviors and potentially also sickness absence such as irregular working hours etc? Lastly, how sickness absence may be prevented in practice is one of the most important questions for many employees. Therefore, the authors could preferably make some recommendations based of these findings. The authors seem to suggest an individually based approach targeting younger individuals with multiple unhealthy behaviors while a vast body of evidence within the work environment research have suggested interventions on an organizational level to decrease sickness absence at the workplace. Furthermore, the authors also point out the importance of combining life-style interventions with improvements in the working environment in the discussion section. What recommendation can be made and how do the knowledge on clustered classes and the use of latent-class analyses support these recommendations? In addition, the percentage of individuals belonging to class 2 and 3 are quite low (17%) – how do this affect the recommendations?
--	--

REVIEWER	Fort, Emmanuel University of Lyon, University of Gustave Eiffel, UMRESTTE, UMR T_9405, F- 69373, LYON, France
REVIEW RETURNED	13-Feb-2023

GENERAL COMMENTS	Figure S1 entitled "Flow chart: selection of the 27 final analytical study sample." is in the main document and in the supplementary material. Author should explain in the discussion part why they used the cut-off of 7/14 units per week for female and male respectively instead of using the same cutoff for both gender. Results present Mean sickness absence days per 10 person-years for 1–7 days' SA 8+ days' SA and all All lengths' SA. it would be interesting to know the prevalence of the population who did not have any SA during the following.
--

	Chi-square tests for comparison of the latent classes of unhealthy behaviours among women and men could be added in the Table S3.
--	---

VERSION 1 – AUTHOR RESPONSE

Reviewer: 1

Dr. Magnus Åkerström, Institute of Stress Medicine

Comments to the Author:

Thank you for the opportunity to review this manuscript.

The study aims to investigate the association between unhealthy behavior and sickness absence among municipality employees of Helsinki using a person-centered approach with latent-class analysis.

The study is well executed, and the manuscript is well written. However, there are some aspects concerning the mechanisms studied and the association between unhealthy behavior, employee sickness absence and the working conditions which warrants some further efforts and discussion. Thank you for your comments on our manuscript. Please find below our point-by-point responses, which we hope to answer your questions and concerns.

Firstly, the authors should preferably discuss the thought mechanism between unhealthy behavior and sickness absence in the introduction and/or the methods section to make the rationale for the study clearer and strengthen the conclusion that sickness absence may be prevented by targeted interventions aiming at increasing unhealthy behaviors. The authors state in the introduction that unhealthy behavior is a major contributor of the employee's sickness absence. It further state that the leading causes for sickness absence is mental and musculoskeletal disorders. What is the direction of these associations? Will unhealthy behavior result directly in sickness absence, or will it result in mental and musculoskeletal disorders which ultimately will result in sickness absence? Or may mental and musculoskeletal disorders, for instance caused by adverse working conditions, result in unhealthy behavior such as low physical activity and alcohol consumption? This need to be clearer to assist the reader in interpreting the outcomes of this study.

Secondly, and in connection to above, the association between unhealthy behavior, employee sickness absence and the working conditions need to be discussed some further. In the introduction, the authors state that unhealthy behavior can mediate the effects of working conditions and socioeconomic circumstances on SA however, but as far as I understand, the references used have shown no effect of working conditions on the association between unhealthy behavior and sickness absence and that working conditions and life-style factors may partly explain the association between education and sick leave.

The manuscript would also benefit from a more detailed description of the professions of the study population and how they correspond to the occupational classes and/or the distribution of professions within the different classes as this may give an increased insight in the working conditions within the different classes that may affect the conclusions of the study.

Thank you for these comments. We have modified the introduction section as suggested and given more detailed examples about the mechanisms and pathways between unhealthy behaviours and sickness absence (p. 3). The focus of this paper was not on explaining the mechanisms between unhealthy behaviours and sickness absence, thus we wanted to keep this short. Additionally, previous studies have not indicated that any of the suggested mechanisms would be superior compared to others (Virtanen et al. 2018, Laaksonen et al. 2009, Quist et al. 2014).

We partially agree with you concerning the references (Robroek et al. 2013 and Laaksonen et al. 2009) and what they claim: Robroek's et al. (2013) study indicated that lifestyle-related factors explained some of the association between education and (10+ days') sickness absence, and

Laaksonen's et al. (2009) study found that working conditions and social class explained some of the associations between health behaviours and sickness absence. Robroek et al. (2013) state: "The results of this study imply that both work-related and lifestyle-related factors do play a role in the mechanisms through which socioeconomic position affects sick leave." In addition, Laaksonen et al. (2009) state: "One possibility is that health-related behaviours are related to sickness absence as mechanisms mediating the effects of stressful working conditions and poor workplace climate on sickness absence". Thus, we think that our statement is correct: "...health behaviours can mediate some of the effects of working conditions and socioeconomic circumstances on SA." (We added a bit more caution with the words 'some of', however).

We have added to the methods section (p. 5) descriptions about the typical occupational titles in different occupational classes as you suggested. As there are hundreds of job titles altogether in the City of Helsinki, we cannot give distributions of professions within different occupational classes. To highlight the diversity in occupations in the study population, we added this sentence to the methods section (p. 3): "The City of Helsinki is the largest employer in Finland with around 38,000 employees and hundreds of occupational titles." Such titles further often have no clear or common translations to English; thus, a more general description is preferred for the readers from different contexts.

In the analyses, the authors control for occupational class which seems to be a measure of socioeconomic position but are there other potential confounder connected to different occupations? Are there factors within the included professions that may affect the measured unhealthy behaviors and potentially also sickness absence such as irregular working hours etc?

The analyses were adjusted for typical confounders that have been used in previous studies on health behaviours and sickness absence. Working conditions and work-related factors (e.g., physical and mental workload, psychosocial working conditions, and working hours) are often considered in this context, but since occupational class is highly correlated with working conditions and we did not want to overadjust the analyses, we left them out from the analyses. Occupational class is a measure of socioeconomic position but also reflects working conditions, thus it was selected instead of education. To select only one measure of work-related factors, such as irregular working hours, should be well reasoned based on the literature. However, we performed sensitivity analysis where we adjusted the models further for mental workload, but this did not contribute to the estimates compared to the current model 2. Finally, our focus was not on factors that explain the associations between health behaviours and sickness absence, and thus, we wanted to keep the models as simple as possible.

Lastly, how sickness absence may be prevented in practice is one of the most important questions for many employees. Therefore, the authors could preferably make some recommendations based of these findings. The authors seem to suggest an individually based approach targeting younger individuals with multiple unhealthy behaviors while a vast body of evidence within the work environment research have suggested interventions on an organizational level to decrease sickness absence at the workplace. Furthermore, the authors also point out the importance of combining life-style interventions with improvements in the working environment in the discussion section. What recommendation can be made and how do the knowledge on clustered classes and the use of latent-class analyses support these recommendations?

Thank you for pointing this out. We may have been somewhat unclear. What we mean with person-oriented approach is not to highlight individual-centred approach to health promotion, but we refer to the method aiming to find latent sub-groups which share similar health behavioural patterns. Thus, the results do not reflect health behaviours at an individual level, but each participant has been assigned to a latent group where they have the highest likelihood to belong to. We focus on these groups of the people that are assumed to follow similar behaviours and how their risk of SA differs from people assigned to another latent group. This approach is increasingly used in social epidemiology, as previous research has been largely variable-oriented. In person-oriented analyses (such as latent class analyses and trajectory analyses), we do not pre-define by ourselves the cut-points or simply, for example, compare all smokers' risk of SA to non-smokers'.

We agree that organisational-level interventions are effective when aiming to reduce sickness absence but so are targeted interventions for groups most at-risk as well. Additionally, organisational-level interventions should pay special attention to occupational groups and employees among whom adhering to healthy behaviours is challenging due to adverse working conditions or socioeconomic circumstances. We have discussed these in the discussion section (p. 12, 3rd paragraph), but we have now revised the paragraph and given more specific examples.

We have added the following sentences: "These could include, for example, reducing physical and psychosocial strenuousness of work while making healthy choices more easily available, for instance, by supporting active commuting, providing exercise facilities, improving availability of staff canteens providing healthy meals, and improving accessibility to occupational health services. Identifying occupational groups among whom these conditions are insufficient and among whom unhealthy behaviours are common is crucial for employers."

In addition, the percentage of individuals belonging to class 2 and 3 are quite low (17%) – how do this affect the recommendations?

We think that the fact that the percentages of individuals belonging to classes 2 and 3 are quite low supports our suggestion for targeted interventions. If the proportion were much higher, organisational-level interventions might be more suitable. Among men, the percentages for class 1 and class 2 were 53% and 47%, correspondingly, which makes it more difficult to apply recommendations for targeted interventions among men. We have mentioned this limitation in the results (p. 9, 1st paragraph) and discussion (p. 13, 1st paragraph) sections.

Reviewer: 2

Mr. Emmanuel Fort, University of Lyon, University of Lyon 1, University of Gustave Eiffel, UMRESTTE, UMR T_9405, F- 69373, LYON, France

Comments to the Author:

Thank you for your comments on our manuscript. Please find below our point-by-point responses, which we hope to answer your questions and concerns.

Figure S1 entitled "Flow chart: selection of the 27 final analytical study sample." is in the main document and in the supplementary material.

Figure S1 is not shown in the main document but only in the supplementary materials. We only mention figure S1 in the methods section (p. 5).

Author should explain in the discussion part why they used the cut-off of 7/14 units per week for female and male respectively instead of using the same cutoff for both gender.

In the methods section (p. 4), we said: "Based on the Finnish recommendations(Finnish nutrition and food recommendations), 7 weekly units for women and 14 weekly units for men were considered as cut-points." We think that referring to the current guidelines is enough to reason our selection of the cut-point used. For women, drinking 7 units or more a week, and for men, drinking 14 unit or more a week, are kept as moderate risk levels for alcohol use. However, we changed the previous reference (Finnish nutrition and food recommendations) to Finnish Current Care Guidelines (<https://www.kaypahoito.fi/hoi50028>), which provides more detailed information on the alcohol consumption recommendations in Finland. The revised sentence is now: "Based on the Finnish Current Care Guidelines on alcohol consumption(Finnish Current Care Guidelines), 7 weekly units for women and 14 weekly units for men (i.e., moderate risk levels) were considered as cut-points."

Results present Mean sickness absence days per 10 person-years for 1–7 days' SA 8+ days' SA and all All lengths' SA. it would be interesting to know the prevalence of the population who did not have any SA during the following.

In the results section (p. 7), we present the proportions of those not having any 1–7 days' and 8+ days' SA during the follow-up: "Of women, 15% had no 1–7 days' SA and 69% had no 8+ days' SA during the follow-up. For men, the corresponding figures were 18% and 75%." We modified the sentence to also indicate the proportions for SA of any length: "Of women, 15% had no 1–7 days' SA, 69% had no 8+ days' SA, and 18% had no SA of any length during the follow-up. For men, the corresponding figures were 18%, and 75% and 17%."

Chi-square tests for comparison of the latent classes of unhealthy behaviours among women and men could be added in the Table S3.

We have added the p-values from Chi-Square Tests to table S3 as requested.

VERSION 2 – REVIEW

REVIEWER	Fort, Emmanuel University of Lyon, University of Lyon 1, University of Gustave Eiffel, UMRESTTE, UMR T_9405, F- 69373, LYON, France
REVIEW RETURNED	27-Mar-2023

GENERAL COMMENTS	I thank the authors for their responses to my past questions and comments. They took in consideration comments made by all the reviewers and they corrected their manuscript. i think this new draft is soon ready to be published. However i have one last question and one last comment. It is not clear if there is no alcohol consumer in this study or if the employees who dont drink alcohol were merged in the "Moderate class" when authors dichotomised the alcohol consumption? It is important to know how many people doesn't drink alcohol, and this result should be added in the text and/or in the table 1. Moreover, a specific class of no alcohol consumer should exist in the latent class analysis. The results of the latent class analysis show that the optimal number of classes were 3 for women and 2 for male (because of a class with only few men with 3 classes). Do these results correlate with the dichotomization of the Health behaviour measures used in the LCA ? Did authors try to anlayses covariates with more classes selected with other LCAs (for exempke with 4 or 5 classes for women and 4 classes for men). Of course the entropy and other statistical criteria for selecting the most optimal number of latent classes were not the best with these choice of number of classes but maybe comparasion of these classes should hav shown other things. Moreover, preventive interventions should have been better to realize with these smallest classes.
---

VERSION 2 – AUTHOR RESPONSE

Reviewer: 2

Comments to the Author:

I thank the authors for their responses to my past questions and comments.

They took in consideration comments made by all the reviewers and they corrected their manuscript. i

think this new draft is soon ready to be published.
 However i have one last question and one last comment.

Thank you for your questions and comments. Please find below our point-by-point responses to them.

It is not clear if there is no alcohol consumer in this study or if the employees who dont drink alcohol were merged in the "Moderate class" when authors dichotomised the alcohol consumption?
 It is important to know how many people doesn't drink alcohol, and this result should be added in the text and/or in the table 1. Moreover, a specific class of no alcohol consumer should exist in the latent class analysis.

The participants who reported not drinking alcohol at all were merged with 'moderate alcohol users'. There were 4% (n=341) of women and 2% (n=51) of men who reported using 0 unit of alcohol per week. In our sensitivity analyses, we found that the associations of weekly alcohol consumption with different lengths of sickness absence were not statistically significantly different between those not drinking alcohol at all and those drinking moderately alcohol. In addition, the numbers of those not drinking alcohol at all were too small to constitute own groups for them. Thus, we decided to merge these two groups.

We added this information to the manuscript, and it now reads as follows (p. 4): "We merged those not drinking alcohol at all (4% of women and 2% of men) with moderate alcohol users since they were few and their associations with SA did not differ from those drinking moderately alcohol."

The results of the latent class analysis show that the optimal number of classes were 3 for women and 2 for male (because of a class with only few men with 3 classes). Do these results correlate with the dichotomization of the Health behaviour measures used in the LCA ?

We were not sure what the reviewer meant with the question, but we hope that our response addresses the reviewer's concern. The main idea of LCA is to identify latent subgroups within a population that share similar characteristics in terms of the selected/included measures. Thus, while individual measures do not necessarily associate with an outcome, the found latent subgroups (i.e., latent classes) including different emphases in the individual measures can be uniquely associated with the outcome. Additionally, LCA is not a method to examine how individual measures correlate with each other (i.e., a variable-based method) but rather identifies a latent variable that explain the patterns of observed values across the cases (Weller et al. 2020). However, we attach here a correlation matrix to illustrate the correlation between the health behaviour measures (blue estimates for men and red estimates for women). As can be seen from the matrix, only weak correlations exist between the measures.

	Leisure-time physical activity	F&V consumption	Sleep sufficiency	Alcohol use	Tobacco use
Leisure-time physical activity	1.00	0.15	0.06	0.07	0.07
F&V consumption	0.10	1.00	-0.01	0.01	0.10
Sleep sufficiency	0.06	0.06	1.00	0.00	0.00

Alcohol use	-0.01	0.07	-0.01	1.00	0.30
Tobacco use	0.03	0.11	0.04	0.32	1.00

We selected the cut-points for dichotomised health behaviour measures based on the literature (e.g., current guidelines) and data (e.g., sufficient group sizes). The used cut-points in the measures may have affected the identified latent classes, and we have mentioned this in the limitations and strengths section in the manuscript. However, since we think that the selected cut-points are well reasoned, we do not think that re-running the analyses with other cut-points for some measures would improve the manuscript.

Did authors try to analyse covariates with more classes selected with other LCAs (for example with 4 or 5 classes for women and 4 classes for men). Of course the entropy and other statistical criteria for selecting the most optimal number of latent classes were not the best with these choices of number of classes but maybe comparison of these classes should have shown other things. Moreover, preventive interventions should have been better to realize with these smallest classes.

We made supplementary analyses where we examined the associations between latent classes of health behaviours and sickness absence using four latent classes for women and three latent classes for men. However, among men in the 3-class model, there were only 39 men in one class, thus the results from the regression analyses are not meaningful to be interpreted. Among women, the 4-class model produced nearly similar classes to the current ones, but the difference was that class 1 (the 'healthiest' one) was basically divided into two classes: one class with overall small probabilities for unhealthy behaviours and another class with small probabilities for unhealthy behaviours except for sleep sufficiency. The associations of these classes with sickness absence revealed that the class with small probabilities for unhealthy behaviours except for sleep sufficiency had increased rates for sickness absence compared to the class with overall small probabilities for unhealthy behaviours. Otherwise, the results resembled the main analyses. Since the statistical model selection criteria supported the selection of the 3-class model for women and the results from the 4-class model did not essentially contribute to the main analyses, we think that the selection of the current model is well reasoned.

VERSION 3 – REVIEW

REVIEWER	Fort, Emmanuel University of Lyon, University of Lyon 1, University of Gustave Eiffel, UMRESTTE, UMR T_9405, F- 69373, LYON, France
REVIEW RETURNED	24-Apr-2023
GENERAL COMMENTS	Dear authors, Thank you for your answer to my last comments and for the modifications made in the draft.